# Chemical and Genotypic Variations in *Aniba rosiodora* from the Brazilian Amazon Forest

**DOI:** 10.3390/molecules26010069

**Published:** 2020-12-25

**Authors:** Diana R. Amazonas, Celso Oliveira, Lauro E. S. Barata, Eric J. Tepe, Massuo J. Kato, Rosa H. V. Mourão, Lydia F. Yamaguchi

**Affiliations:** 1Programa de Pós-Graduação em Recursos Naturais da Amazônia, Universidade Federal do Oeste do Pará, Santarém 68040-255, PA, Brazil; diana.amazonas@gmail.com (D.R.A.); lauroesbarata@gmail.com (L.E.S.B.); 2Institute of Chemistry, University of São Paulo, São Paulo 05508-000, SP, Brazil; celso.chemistry@gmail.com (C.O.); majokato@iq.usp.br (M.J.K.); 3Department of Biological Sciences, University of Cincinnati, Cincinnati, OH 45221, USA; tepeej@ucmail.uc.edu

**Keywords:** rosewood, *Aniba rosiodora*, genotype, chemosystematics, essential oil, metabolic profile

## Abstract

*Aniba rosiodora* has been exploited since the end of the nineteenth century for its essential oil, a valuable ingredient in the perfumery industry. This species occurs mainly in Northern South America, and the morphological similarity among different *Aniba* species often leads to misidentification, which impacts the consistency of products obtained from these plants. Hence, we compared the profiles of volatile organic compounds (essential oils) and non-volatile organic compounds (methanolic extracts) of two populations of *A. rosiodora* from the RESEX and FLONA conservation units, which are separated by the Tapajós River in Western Pará State. The phytochemical profile indicated a substantial difference between the two populations: samples from RESEX present α-phellandrene (22.8%) and linalool (39.6%) in their essential oil composition, while samples from FLONA contain mainly linalool (83.7%). The comparison between phytochemical profiles and phylogenetic data indicates a clear difference, implying genetic distinction between these populations.

## 1. Introduction

Plants of the family Lauraceae have been used remotely in traditional medicine and also to produce flavoring, paper, and timber. Species of this family can be found mainly in tropical and subtropical regions, predominantly in tropical Asia and Central and South America [1]. *Aniba rosiodora* Ducke (also known as Brazilian rosewood), which is frequently spelled incorrectly as *A. rosaeodora* (see Turland et al., 2018, art. 60 [2]), is one of the most economically important species of Lauraceae in Brazil. Its essential oil contains high levels of linalool (76 to 85% of essential oil) [3,4], a valuable element in the perfumery industry. However, in the last two centuries, the intensive exploitation of the bark for essential oil extraction has led to the endangerment of this species.

*Aniba rosiodora* is distributed along the northern part of South America, spreading through Northern Brazil, French Guiana, Guiana, Surinam, Venezuela, and Peru [5]. In Brazil, rosewood occurs in the western region of the Amazonian Basin in the states of Amazonas and Pará, where sparse rosewood forests remain [6,7].

Several studies indicate that the volatile organic compound makeup of rosewood varies from one population to another [8,9,10]. A number of varieties within rosewood have been recognized in the past, but are all currently considered synonyms of a single species, *A. rosiodora* [5]. Due to extreme difficulties in distinguishing specimens based on morphology [7], it is not known whether the phytochemical variants align with the taxonomic varieties. The majority of the studies focused on the chemical and genetic variability in plants from the Amazon State [11,12]. As a further complication, rosewood is frequently misidentified as other species, such as *Aniba parviflora* (Meisn.) Mez, which are similar morphologically, but are not economically important as sources of essential oils [13].

To explore the intraspecific diversity of essential oils among populations from different locations, we collected specimens of *A. rosiodora* from two different areas in the state of Pará: Floresta Nacional do Tapajós (FLONA) and Reserva Extrativista Tapajós-Arapiuns (RESEX). An integrated approach was employed using multivariate analyses to identify the essential oil and non-volatile components that best describe the phytochemical variety in these populations [14,15]. In view of confounding morphological similarity between *Aniba* species, we examined the patterns of phytochemical variation in the context of a molecular phylogeny to understand how the populations of *A. rosiodora* are related to each other and to other *Aniba* species from the area. Thus, the goals of this study were to (1) examine the variation in essential oil composition among populations of *A. rosiodora*, (2) compare non-volatile compounds from leaves, stems, and bark of two *A. rosiodora* populations separated by the Tapajós River, and (3) examine the patterns of essential oil and non-volatile compound compositions in a molecular phylogenetic framework.

## 2. Results

*Aniba rosiodora* collected from the eastern and western sides of the Tapajós River revealed different profiles of essential oil and non-volatile compounds. The hydrodistillation of the aerial parts of plants yielded an average of 2.04% (±0.24) and 0.85% (±0.14) (dry weight) of essential oils for FLONA and RESEX samples, respectively. The volatile composition was also strikingly different for samples collected in each location (Table 1). For example, samples from FLONA contained a relatively low diversity of volatile components, with linalool being the most abundant at 83.7%. In contrast, samples from RESEX presented more evenly distributed components consisting predominately of monoterpenes and sesquiterpenes. The abundance of linalool was only 39.6% in the RESEX samples, followed by α-phellandrene at 22.8%, which was absent in the FLONA samples.

Analysis of the non-volatile chemical profiles by HPLC-ESI-Q-Tof/MS provided further evidence for differentiating between *A. rosiodora* samples from FLONA and RESEX. See Appendix A for base peak chromatogram (BPC) of FLONA and RESEX samples. Principal component analysis (PCA) of these data showed that PC-1 largely explained the variation in these data (Figure 1A). The main compound responsible for differentiating samples from FLONA (RT 7.99 min) had a quasi-molecular ion at m/z 204.0649, calculated for C_11_H_9_NO_3_ [M + H]^+^ 204.0655 and was assigned to the structure of the pyrone anibine (Figure 1B). The fragmentation of this compound exhibits three key ions: *m*/*z* 176.0702 corresponding to the loss of CO, the fragmentary ion at *m*/*z* 144.0440 results from the loss of CO + CH_4_O, and *m*/*z* 172.0388 resulting from the loss of CH_4_O (Figure 2).

The separation of samples from RESEX was due to the presence of 6-styryl-2-pyrone (RT 19.30 min; *m*/*z* 199.0773) (Figure 3), which was previously isolated from *A. parviflora* leaves [16,17]. The fragment ion at *m*/*z* 171.0809 is due to the common loss of CO. The quasi-molecular ion detected in m/z 221.0581 is the sodium adduct of 6-styryl-2-pyrone.

The PCA also indicated the distinction in the chemical profiles of bark samples from other tissues for FLONA samples, described by PC-2 (Figure 1B). This difference is due to the presence of reticuline, a benzylisoquinoline alkaloid in the trunk of *Aniba* species [18]. The isomers of reticuline were identified at RT 3.29 and 2.10 min by *m*/*z* 192.1012, which resulted from the fragmentation of [M + H]^+^ 330.1700 corresponding to reticuline (C_19_H_23_NO_4_; calculated [M + H]^+^ 330.1699). The fragmentations indicated the initial loss of methylamine, forming an ion of m/z 299.1272, followed by fragmentation of the benzyl moiety (*m*/*z* 137.0590). The stable naphthalene ion (*m*/*z* 175.0741) was formed from the rearrangement of the ion *m*/*z* 299. The benzylisoquinoline ion (*m*/*z* 192.1012) was formed directly from the quasi-molecular ion [M + H]^+^, and it was the diagnostic ion for this type of alkaloid (Figure 4) [19].

Since samples of *A. rosiodora* from these two locations presented considerable differences in their volatile organic compounds and methanolic extract composition, samples of *A. parviflora* from Santarém (Pará) and *A. rosiodora* from Presidente Figueiredo (Amazonas) were also analyzed for comparison with the *Aniba* samples from FLONA and RESEX using ^1^H-NMR data. The NMR spectra of FLONA, RESEX, and other samples are in the Appendix A. Analysis of the data was focused on the region between 3.0 and 9.5 ppm to eliminate signals from highly lipophilic compounds (fatty acids). The *A. rosiodora* samples from RESEX were clustered apart from samples of *A. parviflora*, *A. rosiodora* from Presidente Figueiredo, and FLONA, which were all arranged in a distinct cluster (Figure 5). An analysis of the ^1^H-NMR data from crude methanolic extracts reveals complex spectra profiles and the inspection of the loading plot indicates groups of signals differentiating each sample, but the annotation of the compounds was not achievable (Appendix A).

To explore further the relationships among the samples of *Aniba*, a phylogenetic analysis of the samples was performed. The aligned matrices for the plastid *psbA*–*trnH*, *psbD*–*trnT*, *trnC*–*rpoB*, and *trnS*–*trnG* sequences contained 454, 1467, 1057, and 829 characters, respectively, of which 25, 4, and 0 were variable. The aligned, concatenated matrix contained 3807 characters. Of these, a total of 27 (0.8%) characters were variable, and 20 (0.5%) were parsimony informative. All 20 of the parsimony informative characters came from *psbA*–*trnH*.

The topologies of the Bayesian majority rule consensus tree and the maximum parsimony strict consensus of the two most parsimonious trees were not in conflict and differed in only one instances of resolution (Figure 6). *Aniba rosiodora* was resolved as monophyletic with strong support in the Bayesian analysis, but only had moderate support from maximum parsimony. Within *A. rosiodora*, the two accessions from RESEX were supported as a clade that is sister to a moderately supported clade of accessions from FLONA, Presidente Figueiredo, and Manaus. In the analysis, *A. rosiodora* was sister to a clade formed by *A. cinnamomiflora* and *A. hypoglauca*. In all analyses, the samples from RESEX formed a lineage separate from all other *A. rosiodora* specimens (datasets available at Scholar@UC: http://dx.doi.org/doi:10.7945/C21H64).

## 3. Discussion

*Aniba rosiodora* is widespread in the Amazonian basin, but data describing the phytochemical and genetic variation among populations are scarce and most of them focus on Amazonas State specimen variability. In this article, we analyzed two populations from Pará State located in the opposite banks of Tapajós River. This species has long been exploited because the essential oil has a high content of the terpenoid linalool, a compound appreciated by the perfumery industry. The presence of this compound at high concentrations (>30%) is one of the characters that has facilitated the identification of this species [9], since the presence of linalool is rare in the genus *Aniba* [20] and is present in only a few other species (*A. terminalis*, *A. riparia*, and *A. parviflora*; [21,22,23]). The amount of linalool in the essential oils is variable across populations, and in FLONA and RESEX were on average 83% and 39% of the total content, respectively. Not only is the amount of linalool variable between these two populations, but the composition of the essential oils is also variable. Samples from FLONA have a high percentage of linalool and lack phellandrene, whereas the RESEX samples have a much lower linalool content, but had over 22% phellandrene.

The evaluation of the non-volatile compound composition of the leaves, bark, and stems of the samples from FLONA and RESEX indicated a clear distinction between the two populations. The PCA analyses of the mass spectra data pinpointed the pyrone anibine, formerly isolated from *A. duckei* Kosterm. (synon. *A. rosiodora*) [3], as the main compound responsible for differentiating leaves and stem samples from FLONA. Bark samples from the same location presented reticuline, a benzylisoquinoline alkaloid previously identified in the trunk of *Aniba* species [18]. The presence of a pyrone, 6-styryl-2-pyrone, was crucial for the separation of samples from RESEX. This compound was previously isolated from the leaves of *A. parviflora* [16,17]. It is worth noting that both pyrones share a parallel biosynthetic sequence with a common chain elongation step carried out by polyketide synthase-type enzymes, but that differs in the starting units in which nicotinic acid and cinnamic acid are involved to produce anibine and 6-styryl-2-pyrone, respectively (Figure 7). Among the few studies on Pará state samples, *A. rosiodora* from Emílio Goeldi Museum campus (city of Belém), the flavonoid pinocembrin was the major compound identified in the leaves, using mass spectra [9]. Analysis of seedling’s leaves from *A. rosiodora* collected in the Ducke Forest Reserve (Manaus, Amazonas) indicated the benzophenones cotoin and hydroxycotoin as key compounds for these seedlings [11]. Cotoin is found in high amounts in the *A. rosiodora* woods [24], but is not common in leaves. These results indicate that the chemical composition of *A. rosiodora* populations diverge considerably from one another, not only for samples from opposite banks of the Tapajós river, but also for *A. rosiodora* from different locations in the Amazon forest.

The phylogeny of *Aniba* presented here (Figure 6) includes limited sampling (ca. 10% of the species of the genus [5]), but shows several clear patterns. The Brazilian species form a moderately to well-supported clade that is sister to *A. cinnamomiflora* from Costa Rica, Panama, and Venezuela and *A. hypoglauca*, which is restricted to Guyana and Surinam. Among the Brazilian species, the two clades comprising linages of *A. rosiodora*—one including collections from FLONA, Manaus, and Presidente Figueiredo, and the other from RESEX—are supported as separate in all analyses. These results suggest that some of the formerly recognized infraspecific taxa may, in fact, represent discrete lineages with unique phytochemical compositions.

Despite the long history of rosewood exploitation, neither the chemical nor the genetic variability of plants from Pará State classified as *A. rosiodora* has been evaluated. This study shows that considerable dissimilarities exist in essential oil and non-volatile compound composition between two populations morphologically identified as *A. rosiodora*, indicating the presence of different chemotypes in this species. Higher amounts of the linalool were detected in the samples from FLONA, comparing to the RESEX samples that, on the other hand, present α-phellandrene in their essential oil composition. In the compound 6-styryl-2-pyrone, which characterizes the separation of the plants from RESEX, however, anibine and reticuline were distinctive for FLONA samples. The evaluation of phylogenetic patterns provides further evidence for differentiation between the populations of this economically important species.

## 4. Materials and Methods

### 4.1. Plant Material

The specimens were collected in the Brazilian Amazon region from two Conservation Units in the State of Pará: Floresta Nacional do Tapajós—FLONA (S 03°03′13.2″/W 054°58′52.3″) and Reserva Extrativista Tapajós-Arapiuns—RESEX (S 02°30′6.1″/W 055°06′30.6″) (Figure 8). All collections were made under permit 23293-3 from the Sistema de Autorização e Informação em Biodiversidade—SISBIO. Five trees were selected in FLONA with DBH (diameter at breast height) between 27 and 45 cm, and in RESEX, 81 trees were found, but among them, 10 specimens were selected with DBH between 20 and 35 cm. Five hundred grams of each sample were collected in April 2011. The temperature and humidity were 25 °C and 92% in FLONA and 24 °C and 92% in RESEX, respectively, as measured by a portable thermos hygrometer (model ITHT 2220, Instrutemp, São Paulo, SP, Brazil).

The plants from conservation units were identified as *A. rosiodora* by Dr João B. Baitello (Instituto Florestal do Estado de São Paulo) and the vouchers (44874 and 44888, respectively, for RESEX and FLONA) were deposited in Herbário Dom Bento Pickel (Herbarium SPSF) in the Instituto Florestal do Estado de São Paulo, Brazil. Additional samples and species from other locations were analyzed to compare chemical profiles: *A. rosiodora* (voucher Kato 1193) from Presidente Figueiredo, AM (S 2°02’46.2”/W 59°58’05.8”), and *A. parviflora* from Fazenda Curauá, Santarém, PA (S 02°34’15”/W 54°37’08”). These were deposited in the herbarium of the Empresa Brasileira de Pesquisa Agropecuária (EMBRAPA, Herbarium IAN, Belém, PA) (voucher IAN 84897).

### 4.2. Chemical Analysis

#### 4.2.1. Essential Oil Extraction and GC-MS Analysis

One hundred grams of fresh cut aerial parts (leaves and thin branches approximately 6 mm in diameter) of specimens from FLONA (5 individuals) and RESEX (10 individuals) were extracted in triplicate (300 g total), by hydrodistillation for 3 h using a Clevenger apparatus in proportion 1:10 (*w*/*v*). The essential oil percentage was calculated based on the dry weight of the plant material [25], and the humidity of the samples was determined in triplicate using a thermogravimetric moisture analyzer balance (Celtac model DHS-16 A). The average humidity was 49.02% and 58.05% for the FLONA and RESEX samples, respectively. GC-MS analyses were performed using an Agilent instrument (model 6850, Santa Clara, CA, USA) with a DB-5HT capillary column (30 m × 0.25 mm, film thickness 0.10 µm). The initial oven temperature was 66 °C, which was increased by 5 °C/min to 220 °C. The injector and detector temperatures were 250 °C and 230 °C, respectively. The sample was injected using a split ratio of 50:1. The electron impact was set to 70 eV and was monitored in the range of 20 to 500 *m*/*z* with 1.56 scans/s. Retention indexes were calculated relative to C8-C24 n-alkanes, and the identification of essential oil components was based on NIST MS data library version 2.0, followed by comparisons with published literature [26,27,28].

#### 4.2.2. Sample Preparation for Non-Volatile Compounds

Samples of *A. rosiodora* from FLONA and RESEX were collected, and leaves (L), stems (S), and bark (B) were dried in an oven at 40 °C for 24 h and then ground. Powdered samples of each part (200 mg) were extracted with MeOH (HPLC grade) (2 mL) using a crushing disperser (Ultra-Turrax T 25 basic IKA, Staufen, Germany)) for 1 min at 6500 rpm. The extracts were centrifuged at 10,000 rpm for 10 min, and the supernatants were collected for analyses (modified from Matsuda et al., 2011 [29]).

#### 4.2.3. HPLC-ESI-Q-Tof/MS Analyses

The analyses of ElectroSpray Ionization High-Resolution Mass Spectrometry (ESI-HRMS) were performed in a MicrOTOFQ-II Bruker Daltonics mass spectrometer (Q-Tof analyzer, Bremen, Germany) coupled to a Shimadzu HPLC (Kyoto, Japan) system consisting of two pumps LC-20AD, automatic injector SIL-20A, column oven CTO-20A and controller CBM-10A. A Phenomenex Luna 5 µm (150 × 3 mm, 100 Å particle size) column was used. The column oven was kept at 30 °C, and chromatography was performed with a flow of 500 µL/min using MeOH:H_2_O as the mobile phase in a gradient of 0 min 20% of MeOH held until 5 min, from 5 to 30 min 20 to 100% MeOH. The mass spectrometer was operating in electrospray positive mode, with nebulization and drying gas at 4 Bar and 8 L/min, respectively. The capillary voltage was set to 4500 V, and the drying temperature was set to 200 °C. The collision cell and quadrupole energy were set to 12 eV and 6 eV, respectively. Samples were prepared by dissolving 50 µL of the supernatant collected directly from the extracts in 950 µL of MeOH, and filtered (Millex PTFE, 0.45 µm); 10 µL of this sample was injected into the equipment. The HPLC-MS raw data were analyzed by XCMS online software 21, and multivariate analyses were performed using Unscrambler software version 10 (CAMO Analytics, Oslo, Norway).

#### 4.2.4. NMR Analysis

Nuclear magnetic resonance (NMR) analyses were performed using 10 mg of MeOH extract dissolved in 800 µL of CDCl_3_ (99.8% Cambridge Isotopes Laboratories TM) containing 0.05% TMS (tetramethylsilane). The ^1^H-NMR (Bruker DPX 200 MHz) spectra were obtained operating at a hydrogen NMR frequency of 200.13 MHz with a 5 mm probe. The spectra consisted of 256 scans and 300 k data points, with a pulse width of 8.0 µs (30°) and a relaxation delay of 1.0 s. The processing of the spectra was performed in the MestReNova (version 6.0.2-5475) program by automatic Fourier transformation using a line broadening of 0.3 Hz. TMS was used as an internal standard, and the residual hydrogen signal of CDCl_3_ was referenced at 7.26 ppm. The integration of the spectra signals was carried out into regions of equal width (0.05 ppm) of the region δ 3.00–9.50 ppm. The region containing residues of chloroform (7.0 to 7.4 ppm) was excluded from each spectrum. The multivariate analysis was performed using Unscrambler software version 10 (CAMO Analytics).

### 4.3. Phylogenetic Analysis

Five accessions of *A. rosiodora*, including two from RESEX, one from FLONA, one from Presidente Figueiredo, and one from Manaus, as well as *A. hypoglauca* Sandwith and *A. cinnamomiflora* C.K. Allen were included in the phylogenetic analysis. *Ocotea quixos* (Lam.) Kosterm. was included as an outgroup. Voucher information and GenBank accession numbers are listed in Table 2.

Total genomic DNA was isolated from silica gel-dried leaf material using the DNeasy Plant Mini Kit (Qiagen, Inc., Valencia, CA, USA). PCR amplification was carried out using GoTaq Green Master Mix following the manufacturer’s protocol (Promega Corp., Madison, WI, USA). The four molecular markers from the chloroplast genome used were *psbA*–*trhH* [30], *psbD*–*trnT* [31], *trnC*–*rpoB* [32], and *trnS*–*trnG* [33]. After confirming amplification on agarose gels, amplified products were sent to Beckman-Coulter Genomics, Inc. (Danvers, MA, USA) for purification, labeling, and sequencing. Forward and reverse sequences were assembled and aligned in Geneious R11 (https://www.geneious.com).

Bayesian inference analysis was performed using MrBayes 3.2.2 [34,35]. The nucleotide substitution model [36] (F81) was selected using MrModeltest 2.2 [37]. MrBayes was run for 10,000,000 generations, with one tree sampled every 1000 generations and all other settings as the defaults. The first 25% of trees were discarded as burn-in before constructing the majority rule consensus tree and calculating posterior probabilities. Maximum parsimony analyses were performed using PAUP* 4.0a166 [38] using an exhaustive search with default parameters. Bootstrap values for nodes were estimated from a “branch and bound” search of 5000 replicates.

## Figures and Tables

**Figure 1 molecules-26-00069-f001:**
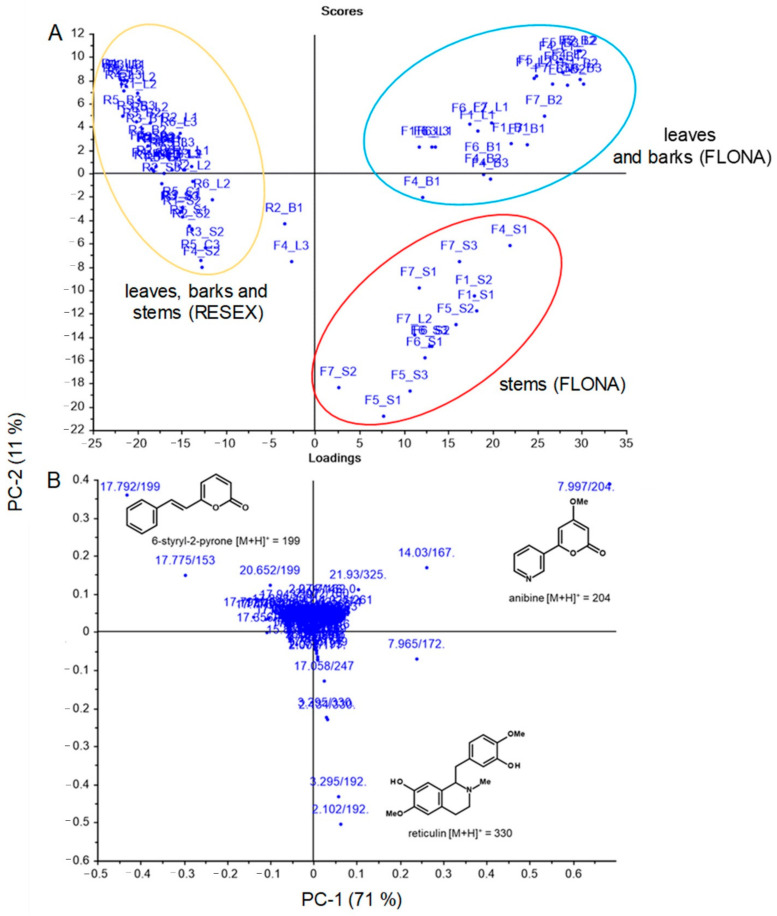
Results of a principal component analysis (PCA) from ESIHRMS data, illustrating the relationship between samples of *A. rosiodora* from FLONA and RESEX reserves. (**A**) The score plot and (**B**) the PCA derived loading plot, presenting the contribution of RT/(*m*/*z*) data with cumulative 82% variance within the two components (PC1 and PC2). Samples are coded by “locality_tissue type”, with R = RESEX, F = FLONA, B = bark, S = stems, and L = leaves.

**Figure 2 molecules-26-00069-f002:**
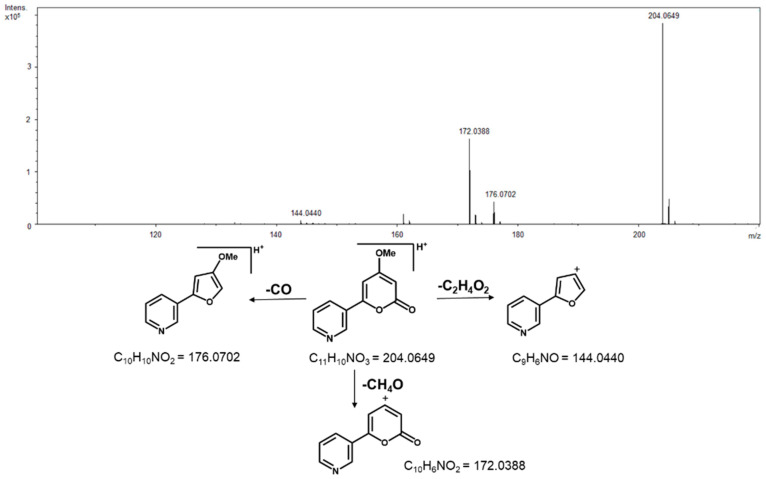
High resolution mass spectrum of anibine [M + H]^+^ = 204.0649 and its fragmentation pathways.

**Figure 3 molecules-26-00069-f003:**
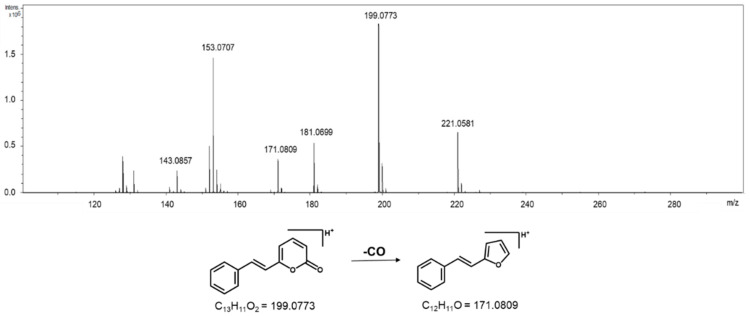
High resolution mass spectrum of 6-styryl-2-pyrone [M + H]^+^ = 199.0773 and its fragmentation pathways.

**Figure 4 molecules-26-00069-f004:**
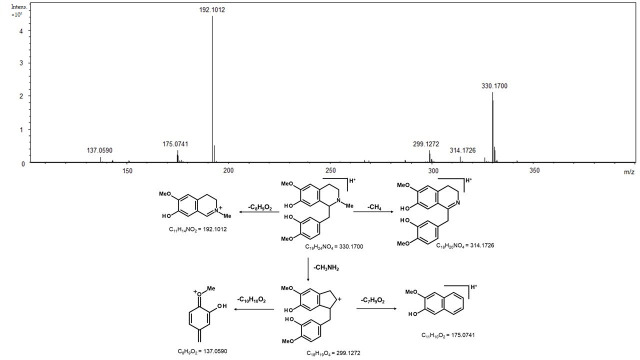
High resolution mass spectrum of reticuline [M + H]^+^ = 330.1700 and its fragmentation pathways.

**Figure 5 molecules-26-00069-f005:**
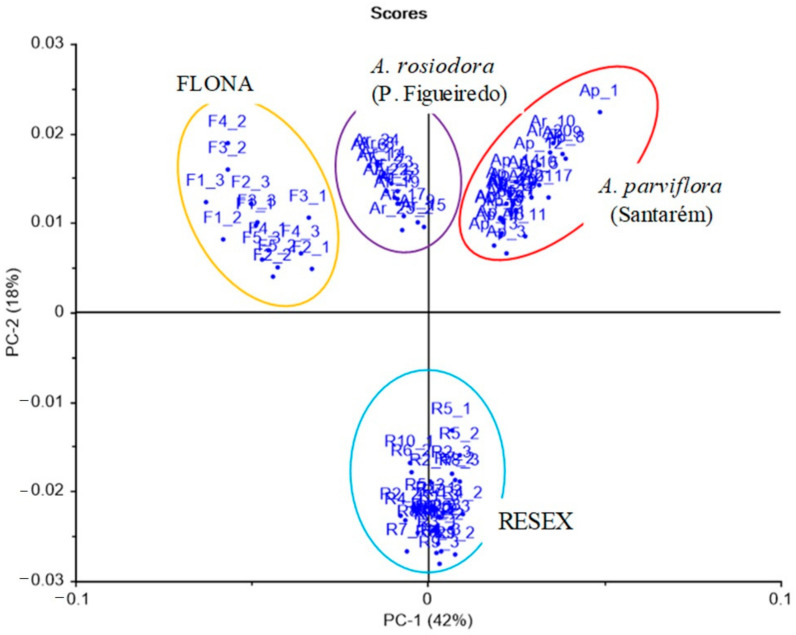
The score plot of the first two PCs of ^1^H-NMR data of *A. rosiodora* samples from RESEX (R), FLONA (F) and Presidente Figueiredo (Ar), and *A. parviflora* (Ap) (Santarém, PA), representing the chemical profile variance in *A. rosiodora* collected in different localities.

**Figure 6 molecules-26-00069-f006:**
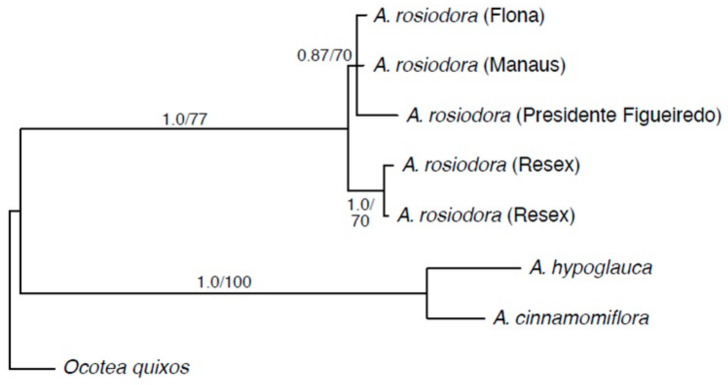
The 50% majority rule tree from Bayesian analysis of a concatenated dataset of plastid *psbA*–*trhH*, *psbD*–*trnT*, *trnC*–*rpoB*, and *trnS*–*trnG*. Branch support values are Bayesian posterior probabilities ≥0.5/maximum parsimony bootstrap ≥50%. The dashed line indicates a branch that was present in the Bayesian topology, but not in the Maximum Parsimony tree.

**Figure 7 molecules-26-00069-f007:**
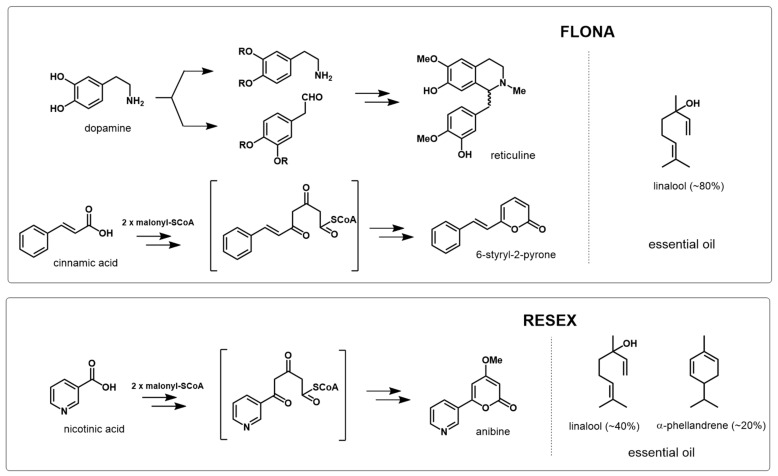
Biosynthetic scheme of the compounds of samples from FLONA and RESEX and the composition of their volatile organic compounds (essential oil).

**Figure 8 molecules-26-00069-f008:**
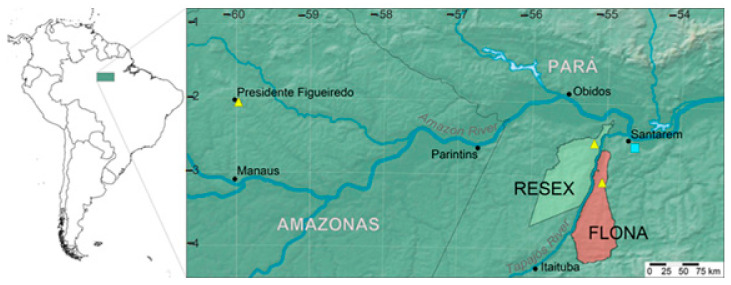
Geographic distribution of the species and samples included in the chemical variation study. The yellow triangles represent *Aniba rosiodora*, and the blue square represents *A. parviflora*. The RESEX (Reserva Extrativista Tapajós Arapiuns) and FLONA (Floresta Nacional do Tapajós) preserves are indicated on the west and east banks of the Tapajós River, respectively.

**Table 1 molecules-26-00069-t001:** Volatiles (%) identified in essential oils from rosewoods from FLONA and RESEX.

Compounds	RT (min)	FLONA (%)			RESEX (%)		
		$\bar{X}$ (5 Individuals)	SD	R.I.	$$\bar{X}$$ (10 Individuals)	SD	R.I.
α-pinene	2.7	nd	-	-	1.7	1.00	931
α-phellandrene	3.6	nd	-	-	22.8	6.76	930
*p*-cymene	3.9	nd	-	-	7.0	1.77	936
β-thujene	4.0	nd	-	-	6.0	3.22	951
β-ocimene	4.4	nd	-	-	2.5	0.81	915
*cis*-linalool oxide	4.8	1.0	0.55	878	nd	-	-
*trans*-linalool oxide	5.1	1.1	0.47	832	nd	-	-
linalool	5.4	83.7	2.63	967	39.6	9.03	962
α-terpineol	7.4	nd	-	-	1.7	0.60	890
γ-elemene	14.7	nd	-	-	3.5	2.08	897
spathulenol	16.5	1.6	0.98	834	3.5	1.07	913
guaiol	17.0	nd	-	-	1.8	0.51	838
*cis* α-santalol	19.5	1.4	0.59	717	nd	-	-
aromadendrene oxide	19.6	2.5	0.86	783	nd	-	-

nd = not detected, R.I. = retention index, SD = standard deviation. – not calculated.

**Table 2 molecules-26-00069-t002:** List of accessions included in the phylogeny analysis in this study, with voucher information (collector followed by code), collection locality, and GenBank accession numbers. Dashes indicate that no sequences were available.

Species	Voucher	Locality	*psbA–trnH*	*psbD–trnT*	*trnC–rpoB*	*trnS–trnG*
*Aniba rosiodora* Ducke	D. Amazonas 22 (SPSF)	BRAZIL. Pará: Floresta Nacional do Tapajós (“FLONA”)	MT679556	MT679562	MT679566	MT679571
*Aniba rosiodora* Ducke	D. Amazonas 226 (SPSF)	BRAZIL. Pará: Reserva Extrativista Tapajós-Arapiuns (“RESEX”)	MT679557	MT679563	MT679567	MT679572
*Aniba rosiodora* Ducke	D. Amazonas 278 (SPSF)	BRAZIL. Pará: Reserva Extrativista Tapajós-Arapiuns (“RESEX”)	MT679558	MT679564	MT679568	MT679573
*Aniba rosiodora* Ducke	Kato 1193	BRAZIL. Amazonas: Presidente Figueiredo	MT679559	MT679565	—	—
*Aniba rosiodora* Ducke	Kato 1446	BRAZIL. Amazonas: Manaus	MT679560	—	MT679569	MT679574
*Aniba cinnamomiflora* C.K. Allen	N. Cuello 955 (MO)	VENEZUELA. Trujillo: Boconó, Parque Nacional Guaramacal	AF268770	—	—	—
*Aniba hypoglauca* Sandwith	A. Chanderbali 165 (MO)	GUYANA. Upper Demerara-Berbice: Iwokrama Reserve	AF268771	—	—	—
*Ocotea quixos* (Lam.) Kosterm.	D. Neill 9487 (MO)	ECUADOR. Napo: Jatun Sacha Biological Reserve	AF261999	—	—	—

## Data Availability

Publicly available gene sequence datasets were analyzed in this study. This data can be found here: https://www.ncbi.nlm.nih.gov/genbank/; accession numbers: BankIt2359259 Seq1 MT679556; BankIt2359259 Seq2 MT679557; BankIt2359259 Seq3 MT679558; BankIt2359259 Seq4 MT679559; BankIt2359259 Seq5 MT679560; BankIt2359259 Seq6 MT679561; BankIt2360320 Seq1 MT679562; BankIt2360320 Seq2 MT679563; BankIt2360320 Seq3 MT679564; BankIt2360320 Seq4 MT679565; BankIt2360328 Seq1 MT679566; BankIt2360328 Seq2 MT679567; BankIt2360328 Seq3 MT679568; BankIt2360328 Seq4 MT679569; BankIt2360328 Seq5 MT679570; BankIt2360340 Seq1 MT679571; BankIt2360340 Seq2 MT679572; BankIt2360340 Seq3 MT679573; BankIt2360340 Seq4 MT679574; BankIt2360340 Seq5 MT679575. The data presented in this study are available in supplementary material.

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
