# Peer review of "Chemical and Genotypic Variations in Aniba rosiodora from the Brazilian Amazon Forest"

_molecules, 2020, doi:10.3390/molecules26010069_

Round 1

Reviewer 1 Report

The present work may be of great interest. The Abstracts and the Introduction are good and show that they have good bibliographic references and good information.
The results should be improved, the authors should improve the information and organize the data they want to expose, for example Table 1 does not exist. On the other hand, they indicate that we should consult the Supplementary Material and this is not available in the Editorial, I have not been able to download this information so I have not been able to verify the corresponding figures.
In Figure 3, a mass spectrum of 6-styryl-2-pyrone is presented with a mass of M+H+ at 199.0773 and a peak appears at the mass value of 221.0581, the question that is placed is to know if this will not be the molecular ion ?.
In writing, a good revision must be carried out, because in line 153 the word "A. sosiodora" must be written in italics.
In Materials and Methods, from lines 248 to line 289 are repeated on the next page, for this reason they should be eliminated.
Discussion can be improved.

Author Response

We would like to acknowledge the kind review and critical advice for improving the manuscript. As requested, the text was revised concerning the English grammar editing. The Supplementary Material was included in the submission and all parts of the Results and Discussion sections were revised for enhance the overall quality of the manuscript.

The difference of the mass between m/z= 199.0773 and m/z = 221.058 of the two quasi-molecular ions are 22 Da, what means that is a sodium adduct formed in the source of spectrometer. This information was added in the results.

“In Materials and Methods, from lines 248 to line 289 are repeated on the next page, for this reason they should be eliminated.”

These information were not excluded because they are complementary to each other.

Reviewer 2 Report

In this manuscript, the chemical and genotypic variations of two different populations of Aniba rosiodora are reported.  The phytochemical profile of the two populations located in different places of the Amazon forest was proposed by comparing the profile of the volatile fraction in terms of volatile organic compounds and the non-volatile organic compounds. The chemical analysis performed include GC-MS experiments to evaluate the volatile fraction and HPLC-MS experiments for the non volatile fraction. The data collected in combination with the PCA analysis of the data allowed to certainly distinguish the different populations. The content of volatile organic compounds was evaluated analyzing leaves, bark and stems providing a clear distinction of the two populations. At the same time, a phylogenetic analysis was also performed confirming the variation of the two populations.  

The topic presented is of interest, considering that Aniba rosiodora is a valuable source of essential oils used in the perfumery industry. The manuscript is well written and well organized. The obtained results were discussed justifying the aim of the study.

I have only some minor comments that may help to improve the quality of the paper:

  • To highlight the comparison between the results obtained in this work with the other types
  • To discuss also the 1H NMR data.
  • Even if the conclusions are not mandatory, it may be of interest for the reader to have a summary of the objectives achieved.

Author Response

We are grateful for the revision and effort made for improve the quality of the manuscript. The English grammar was reviewed by a native English speaker. The results were improved and data from previous studies of A. rosiodora’s phytochemical profiles were added to upgrade the discussion. All the manuscript was extensively revised for enhance the clearness and quality of the manuscript.

Round 2

Reviewer 1 Report

This new version of the manuscript has been completed. The introduction is complete and has been correctly referenced.
All requested changes have been made and texts have been completed.
All questions posed by reviewers have been answered and are consistent with the indicated conclusions.